# Evaluation of EV Storage Buffer for Efficient Preservation of Engineered Extracellular Vesicles

**DOI:** 10.3390/ijms241612841

**Published:** 2023-08-16

**Authors:** Yuki Kawai-Harada, Hanine El Itawi, Hiroaki Komuro, Masako Harada

**Affiliations:** 1Institute for Quantitative Health Science and Engineering (IQ), Michigan State University, East Lansing, MI 48824, USA; haraday2@msu.edu (Y.K.-H.);; 2Department of Biomedical Engineering, Michigan State University, East Lansing, MI 48824, USA

**Keywords:** extracellular vesicles, storage buffer, preservation, engineered extracellular vesicles

## Abstract

Extracellular vesicles (EVs), detectable in all bodily fluids, mediate intercellular communication by transporting molecules between cells. The capacity of EVs to transport molecules between distant organs has drawn interest for clinical applications in diagnostics and therapeutics. Although EVs hold potential for nucleic-acid-based and other molecular therapeutics, the lack of standardized technologies, including isolation, characterization, and storage, leaves many challenges for clinical applications, potentially resulting in misinterpretation of crucial findings. Previously, several groups demonstrated the problems of commonly used storage methods that distort EV integrity. This work aims to evaluate the process to optimize the storage conditions of EVs and then characterize them according to the experimental conditions and the models used previously. Our study reports a highly efficient EV storage condition, focusing on EV capacity to protect their molecular cargo from biological, chemical, and mechanical damage. Compared with commonly used EV storage conditions, our EV storage buffer leads to less size and particle number variation at both 4 °C and −80 °C, enhancing the ability to protect EVs while maintaining targeting functionality.

## 1. Introduction

First discovered in the 1960s [1], extracellular vesicles (EVs) are a heterogeneous population of lipid-bound nanoparticles secreted by all cell types in all species [2]. The role and secretion mechanism were obscure until two separate groups simultaneously reported the selective release of small vesicles containing transferrin receptors that was internalized to reticulocytes in 1983 [3,4]. They revealed the formation of exosomes as intraluminal vesicles of multivesicular endosomes that are secreted extracellularly upon fusion of the endosome and plasma membranes [3]. In the last decades, EV (including exosome) function in cell–cell communication became clearly evident and thus has received increasing attention. Numerous groups reported the function of EVs in a broad range of physiological and pathophysiological processes, including immune responses [5], neurodegenerative diseases [6], inflammatory diseases [7], cardiovascular diseases [8], cancers [9], and infectious diseases [10], which were mediated by transferring functional biomolecules, such as protein, nucleic acids, and metabolites. EVs are an attractive source for diagnostic biomarkers because methods of noninvasive body fluid sampling are mostly well established. In addition, the properties of EVs to protect and transport bioactive molecules are ideal as a delivery vehicle for molecular drugs for the current molecular medicine needing carriers that do not elicit immunogenic responses. However, the progress toward clinical use of EVs is slow due to the shortfall of standardized methods, which often result in experiment-to-experiment variabilities and contradictory findings on their biology and functions, impeding the progress of EV research, not limited to but including isolation, characterization, cargo loading and controlled and mass production of EVs [11]. Therefore, the International Society for Extracellular Vesicles (ISEV) continues to update the essential Information for Studies of Extracellular Vesicles (MISEV) guidelines [12,13,14].

An isolation method is one of the critical processes in EV research and includes differential ultracentrifugation, gradient centrifugation, filtration, affinity immunoprecipitation, and microfluidic isolation. Each method allows the enrichment of slightly different EV populations but is associated with a different grade of impurity, which also depends on the EV source (culture media or bodily fluids) [11,15]. Another area urgently needed in EV research is the physical and chemical properties of storage conditions that influence EV quality for both analyses and downstream applications [16,17]. The most commonly used freeze storage in phosphate buffer saline (PBS) causes damage to EVs during storage leading to the loss of some functions [16,18]. Several groups explored various conditions and the use of additives to improve the conditions, such as the use of Tween [18], bovine serum albumin (BSA) [19], trehalose [20], and DMSO [19], at different temperatures. A recent report by Görgens et al. showed that adding the combination of Human Serum Albumin (HSA) and trehalose to a PBS–HEPES buffer drastically improved EV quality following short- and long-term storage compared to PBS alone [21]. Interestingly, a widely used buffer additive, serum albumin, is the most abundant protein component of the blood of all vertebrates, which has evolved by diversifying the physicochemical, genetic, and physiological biochemical properties, thus variable in different species [22]. While HSA possesses extensively overlapping physicochemical properties, it differs from BSA, for example, in adsorption, crystallization mechanisms, and binding affinity to hydrophilic surfaces [23]. Thus, a careful evaluation of each serum is necessary as these properties influence interactions [24].

In this study, we evaluated the EV storage buffer (EBV), consisting of trehalose and BSA-supplemented PBS–HEPES buffer, compared to the widely used storage methods (PBS or media storage) for the capacity of engineered EVs, focusing on cargo protection, which is one of the critical features in a drug/gene delivery system. Our study complements the previous report by Görgens et al. that used HSA instead of BSA and validates the preservation method using the more readily available BSA, but includes additional parameters [21]. The use of protected EV cargo, rather than the total EVs as the primary independent assessment, can provide a more precise degree of the transportation capacity of EVs within mammalian systems.

## 2. Results

### 2.1. Experimental Design and Generation of Engineered EVs (eEVs) for Storage Test

The experiments were designed to evaluate the short-term storage condition for engineered EVs compared to PBS and culture media as control. Engineered EVs with modified surfaces that package plasmid DNA were subjected to storage using various buffers and temperature conditions for up to 7 days (Table 1). The formulation and condition of the EV storage buffer were carefully selected based on the addition of trehalose [20] and BSA–HEPES [18] reported by Bosch et al. and van de Wakker et al., respectively, which overlaps with the PBS–HAT buffer described by Görgens et al. (Note that the sources of albumin are different).

Engineered EVs were generated as previously described using genetic modification to the HEK293T cells and cultured media isolation by differential ultracentrifugation (Figure 1) [25,26] following the pellet resuspension in each buffer. Fresh samples were subject to each assay within 2 h following the isolation. These EVs were previously characterized and validated for EV surface display and pDNA packaging using NTA, Western blotting, Immuno-TEM, and qPCR [25,26].

### 2.2. EV Storage Condition Influence Recovered Particle Numbers and Sizes

First, we directly observed the size and morphology of eEVs using Immuno-TEM, which showed heterogeneity in size and morphology for each condition (Appendix A). Then, we measured particle concentrations and sizes for each condition using NTA. As shown in Figure 2A–C, PBS-stored eEVs had significant particle reductions over time, while EV buffer-stored eEVs did not show a statistically significant difference after the 7-day storage period. The particle numbers between fresh samples stored in PBS and the EV buffer showed no statistical significance. Storage of EVs without isolation (cultured media) consistently led to the recovery of fewer particles regardless of the storage temperature, suggesting that this storage method caused a more significant loss of particles than isolated EV storage (Figure 2D).

To examine the influence of EV storage on eEV sizes, we compared the diameter of eEVs and found no significant differences in the median peak across samples (Figure 2A–C). However, there was a general trend toward an increased volume of larger particles in PBS or media storage compared to the EV buffer, indicated in the volume/mm3E-6 graph (Appendix A), which was also observed visually in the video capture. Therefore, we further analyzed these data by selecting the particles larger than 200 nm and plotting the volume in percent for each condition, which exhibited a significant increase in large particle volume within the PBS-stored eEVs (Figure 2A). Notably, the difference in the large EV volumes between samples stored in PBS and the EV buffer at −80 °C (Figure 2D) suggests that the EVs in preparation started fusing or aggregating immediately after resuspension in PBS.

### 2.3. EV Storage Conditions Influence the Integrity of eEVs and Packaged DNA Contents

One of the crucial aspects of eEV storage is its influence on the protective properties of eEVs. To address this question, following the comparison of the physical properties of EVs, we isolated pDNAs from eEVs using qPCR before and after the DNase I treatment to examine the change in eEV protective properties. Notably, freezing media (unisolated eEVs) or eEVs stored in PBS reduced EV protection significantly compared to eEVs stored in the EV buffer. In contrast, there was no significant difference in pDNA protection at 4 °C in both buffers (Figure 3A). We further analyzed pDNA copy numbers per EVs based on the qPCR and NTA data and found no substantial differences among storage conditions (Figure 3A and Appendix A). However, when we calculated based on the original media volume by converting to the uL of EV suspension, pDNA copy numbers were reduced both in PBS and −80 °C (Figure 3B). These data implicate total loss of eEVs in PBS and destruction of eEVs by −80 °C storage. Most importantly, the EV buffer retained higher EV numbers and pDNA copies following either 4 °C or −80 °C over the 7-day storage period.

### 2.4. Effect on the eEV Functionality by EV Buffer Storage

Another critical aspect of eEV storage is whether to influence the function as a delivery vehicle, especially its targeting capacity. To test the storage effect, we assessed the binding capacity of eEVs by bioluminescence imaging (BLI) using a previously established system that demonstrated EGFR-targeting EVs [26]. Briefly, EGFR-overexpressing cells were treated with eEVs colabeled with Gaussia luciferase (gLuc) and an EGFR-targeting monobody, washed, substrate added, and binding measured by BLI. As shown in Figure 4, the bioluminescence from bound eEVs was significantly improved in the EV buffer compared to PBS. However, the bioluminescence in the EV buffer was significantly higher (Appendix A), possibly due to the influence of the buffer formulation, suggesting the binding was not enhanced but at least not interfered with. It is supported by the in vivo data in the following.

Further, we used a pancreas-targeted EV display system to assess differences in in vivo function after storing 24 h at 4 °C in PBS or the EV buffer [25]. Briefly, eEVs colabeled with Gaussia luciferase (gLuc) and a pancreas-targeting peptide (p88) were injected into a mouse via tail vein and circulated for an hour. The mouse was sacrificed, and the organs were removed for BLI measurement. As shown in Figure 5A,B, there were no significant differences in targeting capacity or signal intensity, consistent with our previous findings [25], indicating the storage buffer does not influence the targeting capacity of eEVs in vivo.

## 3. Discussion

EVs hold promising potential for diagnostics and therapeutic applications, yet the field lacks standardized fundamental technology, including but not exclusive to isolation methods, characterization, engineering, and functional analyses. The storage of EVs is one such technology that critically affects their integrity, cargo molecule, and functions, influencing the subsequent therapeutic efficacy for clinical applications [27,28]. In this study, we evaluated the effect of short-term storage using buffers containing BSA [19] and trehalose [20] based on previous studies, which complemented the results of the PBS–HAT buffer reported by Görgens et al., who used HSA instead of BSA. Our study demonstrated that EV storage in PBS severely affects the cell culture media-derived bioactive EV integrity, cargo protection, and targeting functions following short-term storage at −80 °C. In addition, the formation of larger particles by particle fusion or aggregation started immediately after resuspension in PBS. We demonstrated that our EV buffer prevented the significant loss of EVs and the formation of larger particles and maintained pDNA packaging capacity after 7-day storage at 4 °C. Notably, engineered EVs displaying targeting molecules preserved their function in vitro when stored in the EV buffer at 4 °C for seven days, where the EV buffer did not alter in vivo target capacity.

There is an urgent need for robust and standardized methodologies because the vast number of EV-related research papers published vary widely in source samples, experimental parameters, procedures, and settings, often not reproducible [29]. Therefore, since the first publication of the minimal information for studies of extracellular vesicles (MISEV), the International Society for Extracellular Vesicles (ISEV) has been making continuous community efforts to recommend reporting requirements [12,13]. For example, the initial condition widely adopted by the community was the isolated EV storage in PBS buffer at −80 °C [15], which has been extensively used, however, without supporting experimental evidence. The recent report comprehensively analyzing the effect of PBS preservations of small EVs at 4 °C, −20 °C, and −80 °C demonstrated that regardless of the temperature or duration, these conditions extensively damaged EVs, and are thus suboptimal [30]. Several studies examined the influence of storage conditions on EVs, including temperature, freeze–thawing, lyophilization [31,32,33], and various buffer conditions [16,20,21]. The use of cryoprotectants, one of the commonly used storage methods for unstable biomaterials, notably improved stored EV quality [16,20,21]. For example, Bosch et al. demonstrated that adding trehalose to PBS prevented aggregation and improved the size distribution and numbers with improved preservation of biological activity [20]. In addition to the direct impact of buffers, EV adsorption to tube walls reduces particle recovery [18]. Precoating storage tubes can reduce the loss due to this adsorption [18]. The use of excipients, such as BSA and Tween 20, further improved EV preservation without the loss of wound healing function [34]. Our data show reduced particle counts and pDNA recovery (Figure 2A and Figure 3C) from PBS EVs, while pDNA per particle was unchanged between PBS and the EV buffer (Figure 3B), supporting the previous report of PBS EV adsorption to tube walls [34]. Note that our data reveal that the adsorption to tube walls occurs immediately after resuspension in PBS to the fresh samples and, thus, should be avoided.

Particle size analysis is the most common and relatively simple analysis method to measure particle sizes and numbers, but current technology has limitations. The widely used particle counters, such as NTA, a widely used method to assess particle counts and size distributions with mean or median peak values, overlook increases in large particles in fewer numbers and smaller-size particles under the detection limit. Our data show that extracted values of larger particles (>200 nm) presented evidence of particle fusion or aggregation within the PBS buffer with statistical means in both fresh and preserved samples (Figure 2B). Yet, these measures lack validation of individual EVs that cannot exclude the possibility of counting non-EV particles, such as transfection complexes [35], protein aggregates, or serum byproducts. Nevertheless, these measures from the current study could serve as one indicator of size variation among easily adaptable storage conditions.

Evaluation of engineered EVs should encompass the essential features for therapeutic applications in addition to physical properties, such as specificity and protection of exogenously introduced cargo after storage. Our data suggest the −80 °C storage in PBS or media led to a loss of the protective function of nucleases, which is consistent with previous reports [16,18,33]. Most of these reports have evaluated the protective functions of EVs by assessing the bioactivity of cargo molecules [36], which is an indirect assessment and cannot rule out the possibility that the molecular complex without EV protection exerts functional activities. To our knowledge, this is the first report evaluating the packaging capacity of eEVs and their changes after storage in PBS or the EV buffer by quantitatively assessing the DNA copy numbers before and after the nuclease treatment. In addition, we demonstrated that our EV buffers retain their eEV protective function and targeting after storage at 4 °C or −80 °C, by in vitro binding assay (Figure 3A–C) and in vivo biodistribution assay (Figure 5A,B). It is worth noting that the stronger bioluminescence signals from the binding assay are likely due to the impact of BSA supplementation and not the higher EV numbers. Previous reports demonstrated the effect of BSA on improved bioluminescence signals [37]. Therefore, our result indicates that the binding capacity is not improved but not disturbed by BSA supplementation.

The limitation of this study is the short-term storage period of up to one week. The data presented here resulted from multiple data points highly reproducible from one experimental condition. Thus, it is critical to evaluate long-term storage using conditions with each experimental variable, such as isolation methods, EV engineering methods, and different source cells. Nevertheless, our particle size parameters obtained from commonly used NTA measurements are widely applicable and potentially serve as one indicator of eEV distortion following storage.

In conclusion, our study provides a comprehensive analysis of the short-term storage effects on media-derived eEVs in PBS and the EV storage buffer on EV size, particle numbers, protective function, and target activity, suggesting an optimal condition for short-term eEV storage. This report presents (1) an eEV size distribution assessment using an overlooked NTA parameter, which led to the finding of the immediate deteriorating effects of PBS storage and freezing on eEVs; (2) the improved storage of our EV buffer; and (3) the method of EV protection assay using quantitative measures of their cargo DNA by qPCR. We believe our work will add evidence and value to standardizing conditions for EV storage, including buffers and temperatures and their analytical methods to generate reproducible research data toward progressing in the field for clinical translation, in addition to the previous efforts for improving EV storage conditions.

## 4. Materials and Methods

### 4.1. Cell Culture and Treatment

The 293T (Human Embryonic Kidney) cell line was obtained from American Type Culture Collection (ATCC) and tested for mycoplasma. The cells were cultured in high-glucose DMEM (Gibco, Waltham, MA, USA) supplemented with 100 U/mL penicillin, 100 µg/mL streptomycin, and 10% (*v*/*v*) fetal bovine serum (FBS, Gibco). Engineered EVs were generated by transfecting EV-display plasmids (pcS-mCherry-C1C2 for storage test, pcS-E626-C1C2/pcDNA-gLuc-C1C2 for in vitro testing, and pcS-p88-C1C2/pcDNA-gLuc-C1C2 for in vivo testing.) into 293T cells using homemade PEI as described previously [25,26]. Following 24 h incubation, cells were washed twice with PBS to remove residual PEI–DNA complex and EVs derived from FBS, and the culture media were replaced with DMEM supplemented with Insulin-Transferrin-Selenium (ITS) Growth Supplement (Corning, Corning, NY, USA), 100 U/mL penicillin, and 100 µg/mL streptomycin for another 24 h incubation for EV production. The clones are available either from the addgene (https://www.addgene.org/Masako_Harada/ (accessed on 1 April 2021)) or the corresponding author upon request.

### 4.2. EV Isolation

The cells were grown in DMEM media supplemented with ITS and Pen-Strep for 24 h, and the media from the plates were collected. For each batch, EVs were purified from 20 mL of conditioned media by differential centrifugation. The media were centrifuged at 600× *g* for 30 min to remove the cells and cell debris. In order to remove the contaminating apoptotic bodies, the media were centrifuged at 2000× *g* for 30 min. The supernatant was then ultracentrifuged in PET Thin-Walled ultracentrifuge tubes (Life Technologies, Carlsbad, CA, USA) at 100,000× *g* with a Sorvall WX+ Ultracentrifuge equipped with an AH-629 rotor (k factor = 242.0) for 90 min at 4 °C to pellet the EVs. The pellet containing EVs was resuspended in 100 µL PBS or the EV buffer. Filter-sterilized EV buffer consisted of 0.2% Bovine Serum Albumin (Life Technologies, Carlsbad, CA, USA), 25 mM D-(+)-Trehalose dihydrate (TCI America, Portland, OR, USA), and 25 mM HEPES pH7.0 (MilliporeSigma, Burlington, MA, USA) in PBS pH7.4 (Life Technologies, Carlsbad, CA, USA).

### 4.3. Nanoparticle Tracking Analysis (NTA)

The particle size and concentration were measured using a ZetaView^®^ (Particle Metrix) Nanoparticle Tracking Analyzer following the manufacturer’s instruction. The following parameters were used for measurement: (Post Acquisition parameters (Min brightness: 22, Max area: 800, Min area: 10, Tracelength: 12, nm/Class: 30, and Classes/Decade: 64)) and Camera control (Sensitivity: 85, Shutter: 250, and Frame Rate: 30)). EVs were diluted in PBS between 20- and 200-fold to obtain a concentration within the recommended measurement range (0.5 × 10^5^ to 10^10^ cm^−3^).

### 4.4. Immuno-Transmission Electron Microscopy (Immuno-TEM)

The EV particle number was determined by NTA for each storage condition. Carbon film-coated 200 mesh copper EM grids were soaked in 50 µL EVs (1 × 10^7^ mCherry EVs in PBS or the EV storage buffer) for 30 min for the adsorption of EVs on the grid. EVs on the grids were fixed by treating with 50 µL of 2% paraformaldehyde (PFA) for 5 min and then rinsed thrice with 100 μL PBS. To quench free aldehyde groups, the grids were treated with 50 μL of 0.05 M glycine for 10 min. The surface of the grids was blocked with a drop of blocking buffer (PBS containing 1% BSA) for 30 min. After blocking, the grids were incubated with 50–100 μL anti-HA (MilliporeSigma, Burlington, MA, USA) or anti-CD63 (Life Technologies, Carlsbad, CA, USA) antibody (1:100 in PBS containing 0.1% BSA) for 1 h. The grids were washed five times with 50 μL PBS containing 0.1% BSA for 10 min each. For secondary antibody treatment, the grids were incubated in a drop of goat anti-mouse IgG coupled with 10 nm gold nanoparticles (Electron Microscopy Sciences, 25512, Hatfield, PA, USA) diluted at 1:100 in PBS containing 0.1% BSA for 1 h. The grids were washed five times with 50 μL PBS containing 0.1% BSA for 10 min each and then with two separate drops of (50 μL) distilled water. EVs were negatively stained with 2% uranyl acetate and then rinsed with PBS. The grids were then air-dried for 24 h, and images were captured by transmission electron microscope (JEOL 1400) at 80 kV.

### 4.5. DNase I Treatment and Plasmid DNA Recovery from EVs

The 2 µL of EVs was incubated at room temperature for 15 min with 1 U of DNase I (Zymo Research, Irvine, CA, USA) and 1× DNA Digestion Buffer. The plasmid DNA was isolated from the EVs using Qiamp Miniprep kits and quantified by qPCR. The plasmid DNA was also extracted from nontreated EVs as a negative control.

### 4.6. Quantitative Real-Time Polymerase Chain Reaction (qPCR)

qPCR was performed using DreamTaq DNA polymerase (Fisher BioReagents). Each reaction contained 200 µM dNTP, 500 nM each of forward (5′-CTAGAGTAAGTAGTTCGCCAGTTAAT-3′)/reverse primer (5′-GCTGAATGAAGCCATACCAAAC-3′), 200 nM probe (5′FAM-ATTGCTACAGGCATCGTGGTGTCA-3′), 0.5 U DreamTaq DNA polymerase, 1× DreamTaq buffer, and 1 µL sample DNA in a total reaction volume of 10 µL using CFX96 Touch Real-Time PCR Detection System (BIO-RAD, Hercules, CA, USA). The PCR amplification cycle was as follows: 95 °C for 2 min, 40 cycles of 95 °C for 20 s, and 65 °C for 30 s. The plasmid DNA (pDNA) copy number was determined by absolute quantification using qPCR to calculate the copy number of EV-encapsulated pDNA per vesicle based on NTA.

### 4.7. In Vitro Bioluminescence Binding Assay

Engineered EVs were generated by cotransfecting EV-display gLuc plasmid (pcDNA-gLuc-C1C2) and EV-display-E626 plasmid (pcS-E626-C1C2) into 293T cells for bioluminescence binding assay as previously described [26]. Particle numbers for each condition were measured right before the binding assay following storage. Briefly, A431 cells were seeded at 2.0 × 10^4^ cells/96-well TC plates 24 h prior to EV treatment. The cells were treated with 2.0 × 10^7^ particles of E626-gLuc-EVs in 100 µL media for 0, 10, 30, and 60 min at 37 °C. Following the three PBS washes to remove unbound EVs, 1 µg/mL coelenterazine-H (CTZ; Regis Technologies, Morton Grove, IL, USA) in 95 µL of DPBS was added to the wells and imaged by an in vivo imaging system (IVIS; Spectrum Perkin Elmer, Waltham, MA, USA). Total photon flux (photons/s) was quantified using Living Image 4.7.2 software (IVIS, PerkinElmer). Values are presented as the means ± SD (*n* = 4).

### 4.8. Animals and Ex Vivo Bioluminescence Biodistribution Assay

Adult female Balb/cJ mice weighting 18–20 g (7–8 weeks old) were used for animal experiments. Animals were purchased from Jackson Laboratories and housed in the University Laboratory Animal Resources Facility at Michigan State University. All the experimental procedures for the animal study were performed with the approval of the Institutional Animal Care and Use Committee of Michigan State University. Isoflurane-anesthetized mice received intravenous injection of 1.0 × 10^10^ p88-gLuc EVs stored for 24 h post-isolation either in DPBS or the EV buffer. Following 60 min circulation, the mice were sacrificed by CO_2_, and the following visceral organs were dissected and placed on a transparent sheet: heart, lungs, liver, kidneys, spleen, and pancreas. Ex vivo bioluminescence images were taken following the application of 200 µL CTZ (10 µg) to each organ, and the total flux (photons/s) of images was quantified using Living Image 4.5 software (IVIS, PerkinElmer). Values are presented as the means ± SD (*n* = 3).

### 4.9. Data Analysis and Statistics

One-way ANOVA and t-test were performed to assess the effect of time and storage conditions. Further details are provided in respective figure legends. Graphs were assembled using GraphPadPrism9 (GraphPad Software version 10.0.1).

## 5. Conclusions

In this study, we evaluated an EV storage buffer that efficiently maintained engineered EVs in concentration, size, and cargo protection for up to one week at 4C and −80C while retaining targeting functionality and physical properties. In addition to previous research, this study provided improved EV storage conditions and additional methods to investigate engineered EV integrity and physical properties critical for EV study and downstream applications. Standardization of EV storage would greatly aid experimental reproducibility in EV research.

## Figures and Tables

**Figure 1 ijms-24-12841-f001:**
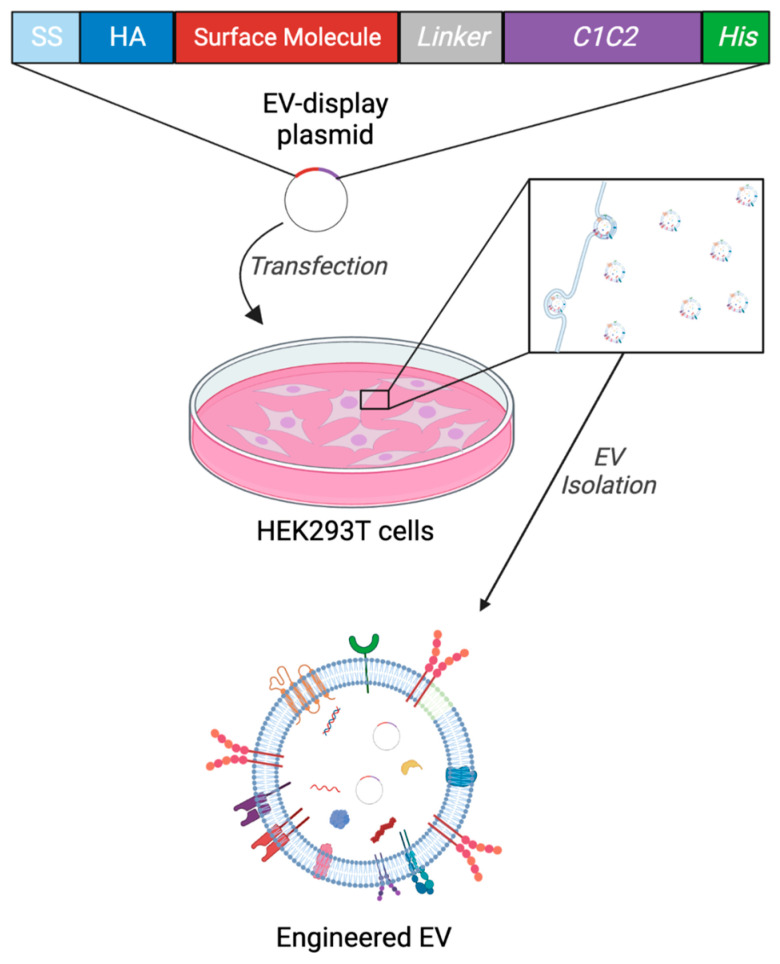
Schematic diagram of the eEV preparation. The EV generation from HEK293T cells by transfecting the EV-display pDNA and secretion of engineered EVs into the cultured media, resulting in the generation of engineered EVs. (Created with BioRender.com: accessed on 1 December 2022).

**Figure 2 ijms-24-12841-f002:**
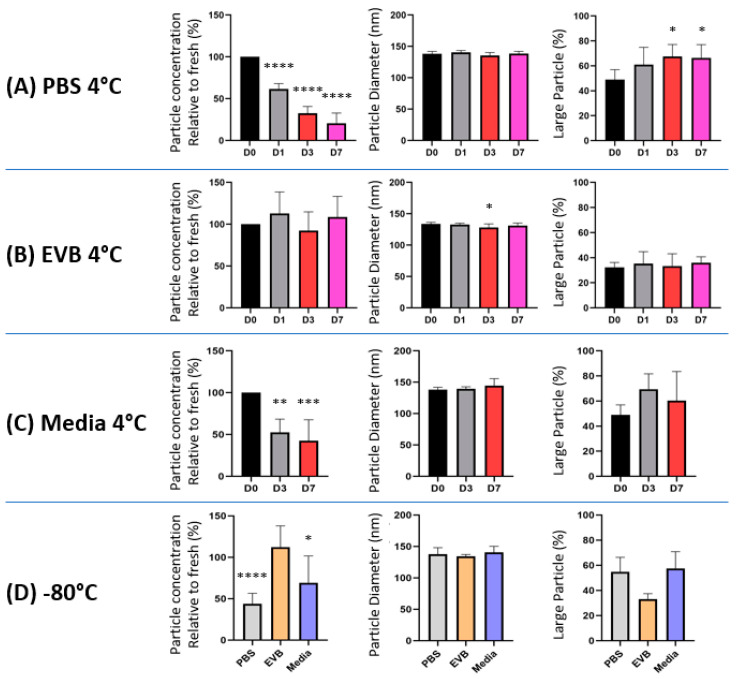
Effects on eEV concentration and sizes after eEV storage in PBS and the EV buffer. eEVs were isolated from HEK293T-conditioned medium by differential ultracentrifugation, resuspended in (**A**) PBS, (**B**) EV buffer, or (**C**) Media without isolation and stored at both 4 °C or (**D**) −80 °C for indicated duration. Effects of storage on EV concentration, size, and volume distribution are shown. Particle number was standardized on yield from 1 mL culture media. The data of PBS Day 0 were used as the data of media day 0. (*n* = 7). One-way ANOVA was used to evaluate the effect of the time course in the group, and unpaired t-test was used to compare different storage conditions. In all figures, significance against Day 0 is expressed as follows: * *p* ≤ 0.05, ** *p* ≤ 0.01, *** *p* ≤ 0.001, and **** *p* ≤ 0001, if not otherwise specified.

**Figure 3 ijms-24-12841-f003:**
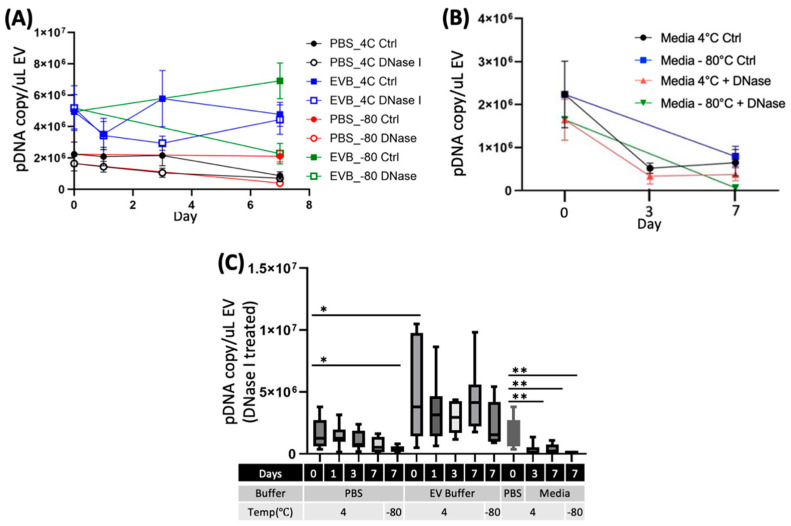
Effects of storage on pDNA protected in eEVs. The introduced pDNA into eEVs were evaluated after storage at both 4 °C or −80 °C for indicated duration. Effects of storage on pDNA copy numbers per 1 µL of EVs are shown, (**A**) PBS and EVB, (**B**) Media, and (**C**) Comparison of encapsulated pDNA in each storage condition. The data of PBS Day 0 were used as the data of media day 0. (*n* = 7). One-way ANOVA was used to evaluate the effect of the time course in the group, and unpaired t-test was used to compare different storage conditions. In all figures, significance is expressed as follows: * *p* ≤ 0.05 and ** *p* ≤ 0.01, if not otherwise specified.

**Figure 4 ijms-24-12841-f004:**
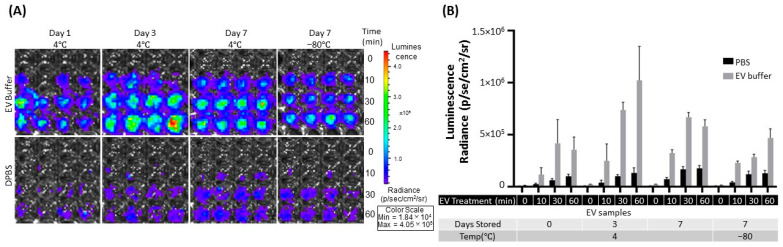
Effects on eEV binding capacity after eEV storage in PBS and the EV buffer. A431 cells (EGFR-positive cells) treated with targeting (E626) EVs. (**A**) Representative image of eEV (gLuc) binding to A431 cells after eEV storage in PBS and the EV buffer for the indicated time. (**B**) The total photon flux (p/s) from EVs bound to the cells by IVIS. The value represents the means ± SD (*n* = 4) in the graph.

**Figure 5 ijms-24-12841-f005:**
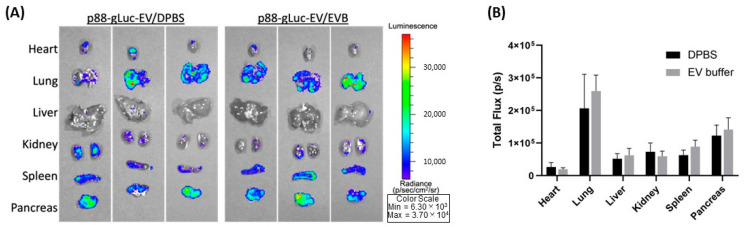
Comparison of biodistribution by peptide-displayed eEVs stored in PBS and the EV buffer. (**A**) Representative images of the organs from the Balb/cJ mice received intravenous injections of p88-gLuc-EVs stored in PBS and the EV buffer. (**B**) The total photon flux (p/s) from EVs bound to the cells was quantified using IVS.

**Table 1 ijms-24-12841-t001:** The evaluated timepoint, storage temperature, and conditions.

Temperature	Buffer	Fresh	Day 1	Day 3	Day 7
4 °C	PBS	+	+	+	+
	EV Buffer	+	+	+	+
	Media (not isolated)			+	+
−80 °C	PBS				+
	EV Buffer				+
	Media (not isolated)				+

+: analyzed time point in this study.

## Data Availability

Data is contained within the article or Appendix A.

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
