# Peer review of "Evaluation of EV Storage Buffer for Efficient Preservation of Engineered Extracellular Vesicles"

_ijms, 2023, doi:10.3390/ijms241612841_

Round 1
Reviewer 1 Report
In the current study, the authors showed that a buffer containing BA, trehalose, and HEPEs, can provide better cargo protection to EVs compared to PBS and culturing media. The authors used different assays to characterize the physical properties of EVs preserved in different conditions and tested the protection functions of the buffer by performing a DNase I treatment.
I have a few questions and suggestions here, which I hope can help improve the quality of the manuscript.
1. My major concern is that only one buffer formulation is tested in the whole manuscript (other than PBS and culture medium, which is more likely to be negative controls). If the current formulation is made based on previous research (ref 21), then I would strongly suggest that the authors add more formulations (original formulation from ref 21 and commercial products) for better comparison.
2. The use of DNase I treatment to stress the EVs is smart and novel to test how buffers can protect cargo molecules. But considering the composition of EVs, I think the RNA and protein content is more abundant than the DNA content. I wonder if the authors have similar “stress tests” for other cargo molecules too.
3. Figure legend for figure 3 is incorrect.
4. For figure 3, please justify why the DNA content went up for some of the groups.
5. For figure 3, as the starting point is different for different groups (different DNA copy numbers per uL), I would suggest the authors also provide a bar graph showing the relative DNA content change as described in the content. (DNA percentage for Day 3 and Day 7 comparing to Day 0 in the same group).
6. For figure 3, no statistic test.
7. Content at line 255-258 should be listed under figure legend?
8. Misalignment of “*” in figure 2.
9. For figure 2 panel D, I would suggest the authors use different colors for the bar representing buffers (to distinguish them from the other panels, where the color represents time points).
10. In the discussion section, I would suggest the authors add more discussion about what the improvement of the current study are comparing to previous studies (especially ref 21), as I did not see too much new information or assays used in the current study.
11. Ref 25 needs to be deleted from the content and list.
12. The use of abbreviation need to be revised (full name still appeared after the abbreviation has been defined).
Author Response
We thank the reviewer for the valuable comments and for listing them in detail.
After carefully considering all the feedback, we revised the manuscript with requested information and figure edits.
The answers to each comment are the following.
- My major concern is that only one buffer formulation is tested in the whole manuscript (other than PBS and culture medium, which is more likely to be negative controls). If the current formulation is made based on previous research (ref 21), then I would strongly suggest that the authors add more formulations (original formulation from ref 21 and commercial products) for better comparison.
We appreciate the reviewer's input. We agree that there are various conditions tested in the previous publications. This study compares the BSA-Trehalose formulation using other parameters rather than broadly to the most commonly used storage method, such as PBS and culture medium, based on previous work, such as ref 18-21. We hope for us and others to perform the comparison with other buffer formulations following this publication to verify our findings.
- The use of DNase I treatment to stress the EVs is smart and novel to test how buffers can protect cargo molecules. But considering the composition of EVs, I think the RNA and protein content is more abundant than the DNA content. I wonder if the authors have similar “stress tests” for other cargo molecules too.
We agree that the RNA and protein content is more abundant than the DNA content in EVs. However, this study evaluated 'exogenously introduced DNA' rather than naturally existing cargo, such as RNAs and proteins, from an EV engineering perspective.
- Figure legend for figure 3 is incorrect.
Addressed accordingly.
- For figure 3, please justify why the DNA content went up for some of the groups.
Addressed accordingly.
- For figure 3, as the starting point is different for different groups (different DNA copy numbers per uL), I would suggest the authors also provide a bar graph showing the relative DNA content change as described in the content. (DNA percentage for Day 3 and Day 7 comparing to Day 0 in the same group).
Bar graph added to the figure.
- For figure 3, no statistic test.
Statistics added to the figure.
- Content at line 255-258 should be listed under figure legend?
Addressed accordingly.
- Misalignment of “*” in figure 2.
Addressed accordingly.
- For figure 2 panel D, I would suggest the authors use different colors for the bar representing buffers (to distinguish them from the other panels, where the color represents time points).
Addressed accordingly.
- In the discussion section, I would suggest the authors add more discussion about what the improvement of the current study are comparing to previous studies (especially ref 21), as I did not see too much new information or assays used in the current study.
We did not intend to improve the previous studies, rather to add analytical approach and validation to the previous study.
- Ref 25 needs to be deleted from the content and list.
Reference modified accordingly.
- The use of abbreviation need to be revised (full name still appeared after the abbreviation has been defined).
The abbreviations were revised accordingly.
Reviewer 2 Report
This is a very interesting paper about the improvement of EV isolation, which is a critical step in the process.
For publication, the author needs minor modifications.
- Without reading, I found formatting problems in the paper. Despite being a minor issue, it means that the author did not read the paper carefully before submitting it. See lines 60-69.
- How was the in vivo experiment mortality rate? Since the animals received intravenous injections.
- What type of anesthesia did you use?
- How was the animal sacrifice method? Please improve the in vivo experiments methodology.
One of the main findings was about the formation of larger particles by particle fusion or aggregation that starts immediately after re-suspension in PBS. The EV buffers prevent EV loss and formation. Very significant in the field.
-
Regarding the EV freezing at -80, was any care taken with the gradual decrease in temperature? Can this interfere with the integrity of the EV?
-
And for thawing the samples, how was the procedure?
-
Do you believe that storage in -20 would have which result? Why did you choose 4 and -80?
Author Response
This is a very interesting paper about the improvement of EV isolation, which is a critical step in the process.
We thank the reviewer for the positive comments.
For publication, the author needs minor modifications.
- Without reading, I found formatting problems in the paper. Despite being a minor issue, it means that the author did not read the paper carefully before submitting it. See lines 60-69.
Addressed accordingly.
- How was the in vivo experiment mortality rate? Since the animals received intravenous injections.
The animal was sacrificed within an hour of injection, thus the mortality rate was not assessed in this study.
- What type of anesthesia did you use?
A type of anesthesia was added to the method section.
- How was the animal sacrifice method? Please improve the in vivo experiments methodology.
Euthanasia method is added to the method section.
One of the main findings was about the formation of larger particles by particle fusion or aggregation that starts immediately after re-suspension in PBS. The EV buffers prevent EV loss and formation. Very significant in the field.
- Regarding the EV freezing at -80, was any care taken with the gradual decrease in temperature? Can this interfere with the integrity of the EV?
- And for thawing the samples, how was the procedure?
These are interesting points and worth further investigation as the duration of temperature change may affect the EV integrity. However, in this work, standard freezing (adding directly to the freezer) and standard thawing were used as a starting point of investigation.
- Do you believe that storage in -20 would have which result? Why did you choose 4 and -80?
We believe the storage temperatures will impact the condition of EVs. -20 storage has risks of exposure to temperature change and mild thawing every time someone opens the freezer, which may result in variability. Thus, we focused on 4C and -80C (non-freeze and freeze) conditions.
Reviewer 3 Report
Abstract: The abstract is written in succinctly where the authors introduce, rationalize and gives a brief of the purpose of their study which highlights the importance of maintaining EVs integrity.
Introduction: The introduction is well written. The second paragraph of the Introduction needs editing Line 60-69 as the text appears to be of different fonts. Please rectify this.
Materials and Methods: qPCR section: The forward and reverse primers sequence has been added properly but the text needs to be justified.
Results and figures:
Fig1: Schematic represents the generation of engineered EVs. This schematic is not very intuitive and does not explain the experimental process in details. The authors can adopt a horizontal template and explain step wise or timeline manner explanation.
Fig.1 should contain the Immuno-TEM images to show the labelling of the engineered EVs under various storage conditions. The NTA analysis can also be included in main figures. These are the characteristic assays that identify EVs which serves as important parameters to rationalize the goal of this study.
Fig.2: The representation of the graphs can be improved. The statistical analysis can include arrows indicating the control vs comparator group. Also, the text font can be reduced and maintained constant.
Fig.4: There must be statistical significance in the data PBS vs EV storage buffer. In the corresponding result section how do the authors conclude that this signal is not due to the EV storage buffer formulation?
Fig.5: Check for spelling of IVIS. Did the authors find any statistical significance of EVs distribution PDS vs EV buffer in various organs, please specify which test has been used? The in vivo data does not show any significance difference of eEV PBS vs EVs storage buffer.
Minor edits to the English language are required.
Author Response
Abstract: The abstract is written in succinctly where the authors introduce, rationalize and gives a brief of the purpose of their study which highlights the importance of maintaining EVs integrity.
We thank the reviewer for the positive comments.
Introduction: The introduction is well written. The second paragraph of the Introduction needs editing Line 60-69 as the text appears to be of different fonts. Please rectify this.
The font was changed to fit the format.
Materials and Methods: qPCR section: The forward and reverse primers sequence has been added properly but the text needs to be justified.
The text was justified.
Results and figures:
Fig1: Schematic represents the generation of engineered EVs. This schematic is not very intuitive and does not explain the experimental process in details. The authors can adopt a horizontal template and explain step wise or timeline manner explanation.
This figure briefly explains how we generated engineered EVs, but not to explain the experimental detail within the figure.
Fig.1 should contain the Immuno-TEM images to show the labelling of the engineered EVs under various storage conditions. The NTA analysis can also be included in main figures. These are the characteristic assays that identify EVs which serves as important parameters to rationalize the goal of this study.
We included Immuno-TEM and NTA analyses in the supplemental figures. The characterization of these EVs was performed and presented in our previous publications. We have run extensive analyses on both Immuno-TEM and NTA, where these widely used analytical methods did not lead to the detection of EV alterations, possibly due to the processes for sample preparation for TEM affecting the morphological characteristics, and the general NTA parameters, which was not sufficient to detect differences. Therefore, we believe that these data need less attention than the ones presented in the main figures.
Fig.2: The representation of the graphs can be improved. The statistical analysis can include arrows indicating the control vs comparator group. Also, the text font can be reduced and maintained constant.
Addressed accordingly.
Fig.4: There must be statistical significance in the data PBS vs EV storage buffer. In the corresponding result section how do the authors conclude that this signal is not due to the EV storage buffer formulation?
The figure revealed that the EV storage buffer does not change the binding capacity of engineered EVs. We provided evidence of this EGFR-targeting EVs with high affinity to EGFR-overexpressing cells. The bioluminescence signal is unique to EV-bound gaussia luciferase, well-described in our previous publication (ref 26). As the amount of EV buffer is consistent across wells, it is hard to believe the EV storage buffer formulation generates a gradient pattern of bioluminescence across various incubation times.
Fig.5: Check for spelling of IVIS. Did the authors find any statistical significance of EVs distribution PDS vs EV buffer in various organs, please specify which test has been used? The in vivo data does not show any significance difference of eEV PBS vs EVs storage buffer.
This figure indicates EV buffer does not show any differences in the biodistribution, as compared to EVs stored in PBS, indicating that the EV storage buffer did not alter the binding capacity of the engineered EVs and thus has no significant differences.
Round 2
Reviewer 1 Report
I appreciate the authors for addressing all my questions and concerns.
The revised manuscript looks good to me with some minor formatting issues:
1. line 116, cm-3;
2. letter "p" should be italic at figure 2 and 3 legend;
3. line 251, should be" -80°C" instead of "80 °C"
4. In figure 3, I do not see any "***" and "****" in the figure, I would delete the corrosponding explanation of "***" and "****" in the legend.
5. The use of abbreviation still need to be revised (full name of BSA was given at line 339, instead of where it first appeared)
Author Response
We thank the reviewer for the detailed suggestions and addressed the comments line by line.1. line 116, cm-3;
Addressed accordingly.
2. letter "p" should be italic at figure 2 and 3 legend;
Addressed accordingly.
3. line 251, should be" -80°C" instead of "80 °C"
Addressed accordingly.
4. In figure 3, I do not see any "***" and "****" in the figure, I would delete the corrosponding explanation of "***" and "****" in the legend.
Addressed accordingly.
5. The use of abbreviation still need to be revised (full name of BSA was given at line 339, instead of where it first appeared)
We noted that the first full name of BSA appeared in the introduction and changed the following accordingly.